# The Consequences of Budding versus Binary Fission on Adaptation and Aging in Primitive Multicellularity

**DOI:** 10.3390/genes12050661

**Published:** 2021-04-28

**Authors:** Hanna Isaksson, Peter L. Conlin, Ben Kerr, William C. Ratcliff, Eric Libby

**Affiliations:** 1Department of Mathematics and Mathematical Statistics, Umeå University, 90187 Umeå, Sweden; hanna.isaksson@umu.se; 2Integrated Science Lab, Umeå University, 90187 Umeå, Sweden; 3Georgia Institute of Technology, School of Biological Sciences, Atlanta, GA 30332, USA; peterlconlin@gmail.com (P.L.C.); william.ratcliff@biology.gatech.edu (W.C.R.); 4Department of Biology, BEACON Center for the Study of Evolution in Action, University of Washington, Seattle, WA 98195, USA; kerrb@u.washington.edu; 5Santa Fe Institute, Santa Fe, NM 87501, USA

**Keywords:** binary fission, budding, multicellularity, aging, adaptation, filaments

## Abstract

Early multicellular organisms must gain adaptations to outcompete their unicellular ancestors, as well as other multicellular lineages. The tempo and mode of multicellular adaptation is influenced by many factors including the traits of individual cells. We consider how a fundamental aspect of cells, whether they reproduce via binary fission or budding, can affect the rate of adaptation in primitive multicellularity. We use mathematical models to study the spread of beneficial, growth rate mutations in unicellular populations and populations of multicellular filaments reproducing via binary fission or budding. Comparing populations once they reach carrying capacity, we find that the spread of mutations in multicellular budding populations is qualitatively distinct from the other populations and in general slower. Since budding and binary fission distribute age-accumulated damage differently, we consider the effects of cellular senescence. When growth rate decreases with cell age, we find that beneficial mutations can spread significantly faster in a multicellular budding population than its corresponding unicellular population or a population reproducing via binary fission. Our results demonstrate that basic aspects of the cell cycle can give rise to different rates of adaptation in multicellular organisms.

## 1. Introduction

The evolution of multicellularity gave rise to organisms whose scale and complexity significantly exceed those of their unicellular ancestors [1,2]. Yet, the early stages of multicellularity were likely primitive and precarious owing to the small genetic distance from unicellularity [3,4,5,6,7,8]. Central to the success of any nascent multicellular organism was the ability to gain adaptations and outcompete its ancestors along with other unicellular and multicellular lineages present in the environment [9]. Although many factors influence the ability of a multicellular lineage to gain adaptations—e.g., physical structure [10,11,12], the multicellular life cycle [13,14,15,16], and eco-environmental conditions [17,18,19]—a crucial factor may be the suite of initial traits of the component cells [20]. Comparative genomics studies show that co-option of ancestral unicellular genes is among the most common modes of adaptation in diverse multicellular lineages [21,22,23,24,25]. In some cases, the existence of certain characteristics in unicellular ancestors may potentiate the evolution of novel multicellular phenotypes. For example, it has been proposed that the existence of temporally alternating phenotypes found in the unicellular ancestors of metazoans, dictyostelid social amoebas, and the volvocine algae were co-opted for spatial cellular differentiation during the evolution of multicellularity [22,26,27,28]. Thus, traits that evolved first in unicellular organisms can influence the tempo and mode of multicellular evolution in that lineage.

Broadly speaking, prior work examining adaptation in simple multicellular organisms has focused primarily on the mode, not the tempo of adaptation. Many of the major conceptual topics surrounding the evolution of multicellularity fall under this category. These include: (i) how multicellular groups form and become Darwinian units capable of adaptation [18,29,30,31,32,33], (ii) how the transition to multicellularity affects subsequent evolutionary processes [6,34], (iii) how cooperative behaviors associated with complex multicellular organisms (e.g., cellular differentiation) evolve and remain stable in the face of social defection [35,36,37,38,39,40], (iv) how multicellular life cycles arise and shape the subsequent evolution of multicellularity [13,14,41], and (v) how multicellular lineages co-opt and modify traits of their unicellular ancestor for novel multicellular purposes [21,27,42,43,44].

Despite the significant body of research on the mode of adaptation in early multicellularity, little attention has been given to the tempo of multicellular adaptation. Like their unicellular ancestors, multicellular organisms need to fix beneficial mutations and purge deleterious ones. The few studies on the tempo of multicellular adaptation have compared different multicellular life cycles with an aim towards understanding variation and competitive differences between life cycles [13,16]. These studies have not considered how different unicellular life history traits can affect the rate of multicellular adaptation. Here we consider how a trait fundamental to unicellular life, whether it reproduces via budding or binary fission, can have consequences for multicellular adaptation.

Budding and binary fission are two common modes of reproduction in single-celled organisms. With budding, a parent cell creates an outgrowth that eventually becomes a daughter cell. With binary fission, a parent cell reproduces by splitting in half. Both types of reproduction are common among unicellular organisms [45], and can be found within extant multicellular taxa. For example, multicellular cyanobacteria [46] typically grow via binary fission, as do plant and animal cells [47], and filamentous fungi grow via budding [48]. Although both mechanisms of reproduction give rise to two cells, they are not identical. Budding creates a clear parent/offspring distinction in which age-related damage, e.g., oxidatively damaged proteins and protein aggregates, is split asymmetrically between parent and daughter cells (see review [49]). In contrast, binary fission partitions components more symmetrically which is readily observable in the age of the cell poles of the two resulting cells (the age of cell poles correlates with age-related damage [50]). Since aging can correspond to a decrease in the rate of reproduction, it affects the relative rates that cells in a population reproduce and thus the rate that a new mutant may spread.

Another difference between budding and binary fission can manifest when populations evolve to be multicellular. Single-celled organisms can form groups—a basic requirement for multicellularity—in two main ways. They can ‘come together’ via aggregation or ‘stay together’ by failing to separate after reproduction [51]. In this paper, we focus on multicellular organisms that form by ‘staying together’ because it is a common mode of multicellular development [45]; it favors the evolution of cooperation [52] and cellular differentiation [37]; and all ‘complex’ multicellular organisms (i.e., animals, plants, red algae, brown algae, and fungi) grow through cell adhesion following cell reproduction [53]. If multicellularity evolves via cells staying together, then the physical connection between cells must be maintained, at least until the multicellular organism fragments. Cells can still reproduce via binary fission while being connected since the axis of division is down the middle; however, in budding there needs to be space for a new outgrowth else the budding cell may not have room to reproduce [11,12]. So, in the same multicellular structure (see Figure 1 for a filament) whether cells reproduce via binary fission or budding may determine which cells reproduce.

Here, we examine how the ancestral cell’s mechanism of reproduction affects the temporal dynamics of adaptation when organisms grow in a simple and mathematically-tractible multicellular form, a filament. We consider a mutation that increases cellular growth rates and use a mathematical model of population growth to compare how quickly the mutation spreads in unicellular and multicellular populations that reproduce via budding or binary fission. Our theoretical approach allows us to disentangle how two key differences between budding and binary fission, i.e., which cells reproduce and how aging propagates, affect population adaptation. Ultimately, we find that in aging unicellular populations, binary fission leads to faster adaptation, while in multicellular populations budding can either inhibit or significantly increase rates of adaptation depending on which cell mutates. Our results show how a fundamental aspect of a cell can affect the rate of evolution in multicellular organisms.

## 2. Materials and Methods

### 2.1. Population Growth and Reproduction

We simulate the population dynamics of multicellular filaments using a deterministic model of population growth. All cells belong to haploid multicellular filaments which have a regular life cycle that reproduces via fragmentation. Filaments grow until they reach a set size for fragmentation, *N* cells (10 in most simulations), at which point they fragment evenly into two daughter filaments, each of size N/2. If cells reproduce via binary fission then every cell in a filament can reproduce. In contrast, if cells reproduce via budding then only cells at the terminal ends of filaments can reproduce. Thus, we continually track the position of cells within filaments to determine if they are able to reproduce. Simulations begin with a single cell that reproduces every 1 time unit. We iterate via discrete time steps (each of duration $10^{-2}$, where the fine temporal resolution is required to accommodate different rates of cell reproduction due to aging) until the total population reaches $10^{5}$ cells. If at the last time step the number of reproducing cells would result in a population that surpasses the carrying capacity, then a random sample of cells is chosen for reproduction so as to reach the carrying capacity exactly.

### 2.2. Mutations

We introduce a single mutation when the population reaches $10^{3}$ cells. We randomly choose one cell to mutate either out of the pool of all available cells or those cells reproducing during the current time step. The difference between these choices reflects differences in how mutations occur, mechanistically. The choice of any cell corresponds to a mutation that occurs due to spontaneous damage while the choice of reproducing cells corresponds to a mutation that occurs due to faulty base incorporation during reproduction. Mutations that occur during cellular reproduction will segregate with equal probability to the parent or the daughter cell (in budding) or either daughter (in binary fission). We note that in the case of a mutation occurring in all available cells, there is the possibility a cell that is just about to reproduce but has not yet done so will gain the mutation and give an immediate boost to the mutant proportion. In all simulations in the main text, the only difference between mutant cells and ancestral cells is in the growth rates, mutants reproduce twice as fast (i.e., every 0.5 time units). In Appendix C, we consider the effects of different mutant growth rates.

### 2.3. Aging

To include the effects of aging we track the time that cells are created (in budding populations) and poles are created (in binary fission populations). In budding populations whenever a cell reproduces, its daughter cell starts anew such that its creation time is the current time step. In populations undergoing binary fission, when a cell reproduces it produces two daughters. Each daughter gets one of the parent’s old poles in addition to a new pole. The point of contact between the two daughters includes the new poles while the old poles are on opposite sides (see Figure 1).

In simulations with a fitness cost to aging, older cells reproduce more slowly than newer cells. We impose the fitness cost by adjusting the reproductive time for cells based on their age. So if a cell is eligible to reproduce we calculate the next time it reproduces by scaling its baseline reproductive time (1 for ancestral cells and 0.5 for mutants) by a function of age f(a):(1)$$f\left(a\right)=b^{a},$$
where *a* is the chronological age of the cell and *b* is the aging factor—a value in the range [1,1.25] that controls the rate at which reproductive time increases. We note that an aging factor of b=1 represents the case in which there is no decline in growth rate due to cell age. The age *a* of budding cells is the difference between the current time and the time they were created, and the age *a* of cells reproducing via binary fission is the difference between the current time and the age of the oldest pole. In Appendix B, we consider the effects of defining the age *a* of cells reproducing via binary fission as the average pole age.

### 2.4. Resource Sharing

In our model, multicellular budding populations reach carrying capacity at a later time than other populations because of the limitations restricting which cells can reproduce. To align populations so that there is no difference in the time it takes to reach carrying capacity, we assume that multicellular filaments can share resources so that non-reproducing cells can contribute resources to reproducing cells and increase their growth rate—such behavior occurs in some multicellular fungi [54,55]. We implement resource sharing by a scaling factor similar to the aging factor f(a). Here the scaling factor h(n) is a decreasing function of the number of non-reproductive cells *n* in the filament such that:(2)$$h\left(n\right)=\frac{1}{k_{0}n},$$
where $k_{0}$ is a constant. We compute $k_{0}$ so that multicellular budding populations, without mutants or age-related changes in growth rate, reached carrying capacity at the same time as unicellular populations.

### 2.5. Noise

For simulations in which noise was added to cell reproduction times, we compute the next time for reproduction after implementing any scaling due to age and then add a random number sampled from a normal distribution with mean of 0 and standard deviation of 0.01 or 0.05. The resulting reproduction time is then rounded to the nearest 0.01 to maintain the discrete time steps of the simulations.

### 2.6. Code

All simulations and computational analyses were conducted in MATLAB R2020b.

## 3. Results

We explore the effects of budding versus binary fission using a simple model of multicellularity. We assume that cells are members of multicellular filaments that have a regular life cycle: cells in filaments reproduce until the filament reaches a set number of cells (10 for most simulations) at which point it splits in the middle to produce two equally sized daughter filaments (5 cells in each). In this model there is no explicit advantage to being in a multicellular filament; rather we assume that some selective pressure has led to, and maintains, multicellular filaments.

The mechanism of cell reproduction, either binary fission or budding, has two immediate consequences in our model. The first concerns which cells in a filament can reproduce. For cells reproducing via binary fission there is no limitation so the resulting population dynamics are the same as if the population were strictly unicellular. In contrast, if cells reproduce via budding we assume that only cells on the terminal ends of a filament can produce daughter cells. The reasoning behind this is that once a cell has daughters on two opposite sides, there is no room for a new daughter without violating the filament structure. Thus, the terminal ends of filaments are the only areas of active reproduction.

The second consequence of whether cells reproduce via budding or binary fission concerns the age structure of the filament, and by extension, population. For cells reproducing via budding there is a definitive parent-daughter relationship such that the bud becomes the younger cell. In contrast, cells reproducing via binary fission do not impose such a distinct partitioning of cell age between the products of reproduction. Rather, cells reproduce by splitting down the middle, so that each resulting daughter cell will have an old pole from the original cell and a new pole resulting from cell division. The age of the resulting daughters will then be some function of the two poles. For our model we consider age to be the maximum pole age though we also show some results if age is instead defined as the average of the two poles, which is proportional to maximum pole age.

Figure 2 shows the age distribution within filaments after repeated rounds of growth and fragmentation. In filaments where cells reproduce via binary fission there is a characteristic pattern that alternates between old and new cells (see Figure 2A). The oldest cells—those with the oldest poles—are at the ends of the filament, and the next oldest cells are in the middle. When the filament fragments, each daughter filament has one of the same terminal cells as the parent and the other terminal cell was previously in the middle of the parent fragment, thus the ends of daughter filaments continue to be older cells. The regular structure shown in Figure 2A, however, relies on cells reproducing at the same rate and filaments having $2^{k}$ cells where *k* is an integer. Figure 2C shows examples of filaments with 10 cells instead of 8, in which some cells have reproduced an extra time compared to others. Although the structure is less regular the oldest cells still tend to be at the ends.

In the case of filaments formed by budding cells, both 8 cell and 10 cell filaments show a similar pattern to each other (see Figure 2B,D): the youngest cells are at the ends of the filament and the oldest cells are towards the middle. There are also sequences of 3 or more cells that differ by a single generation and are monotonically increasing/decreasing in age. Such patterns result from the direct parent-daughter relationship of budding and are not found in binary fission. The spikes in cell age across a filament indicate previous fragmentation events. When a budding filament fragments, one of the inner cells that was previously prevented from reproduction can now reproduce again. This causes spikes in the distribution of ages across a filament. If a filament were started from a single cell and grown to the size immediately prior to fragmentation then the resulting age pattern would look like an inverted letter V without any spikes.

To discern the effects of reproductive mode on multicellular adaptation, we first consider populations in which aging does not alter the growth rate of cells. We begin populations with a single cell and expand them to $10^{3}$ cells, at which point we mutate a randomly chosen member of the population. We examine two possibilities: (1) the mutation can happen in any cell and (2) the mutation can happen in only reproducing cells. The difference between these scenarios stems from the nature of how mutations arise in populations, either they occur randomly with some fixed probability in time or they occur as a result of DNA copying mistakes when cells reproduce. In either case, the mutated cell has a growth rate advantage so that it reproduces twice as fast as the rest of the population—we chose a significant growth advantage to better observe the spread of the mutation within a single population expansion. After introducing the mutant, the population continues growing until it reaches $10^{5}$ cells. We calculate the mutant fraction of the $10^{5}$ cells and use it as a proxy for the rate of adaptation of the multicellular population by dividing it by the starting fraction, i.e., $10^{-3}$. Since each simulation involves randomness in terms of which cell becomes the mutant, we run 1000 simulations to obtain a distribution of mutant proportions.

The distributions of mutant proportions differ between budding populations and those reproducing via binary fission (see Figure 3). In the case of binary fission, all cells reproduce so the choice of the initial mutant cell is mostly inconsequential—except for the rare case (≈2%) in which a cell is mutated immediately before it is set to reproduce. The resulting distribution in Figure 3a has a single peak with the mutant proportion increasing by an average factor, or fold, of 111.47±4.91. For the case of budding populations, Figure 3B shows that the final mutant proportion is more sensitive to the particular cell mutated. This sensitivity comes from the fact that the expected time for cells to reproduce depends on their positions in multicellular filaments. Terminal cells are the only ones that reproduce and so if one of those mutates then the mutant will reproduce again in 0.5 time units. If instead, the mutation occurs in a cell in the middle of the filament it might need to wait several generations before it can reproduce. This causes the distribution to have a greater dispersion than the one for binary fission populations.

The mechanism of mutation also has an effect in budding populations. If the mutation can occur in any cell then there are many different waiting times until the mutant will get a chance to reproduce. In contrast, if mutations can only happen in reproducing cells then they must occur in a terminal cell as it is budding. The mutation then has a 50% chance of occurring in the daughter which will reproduce again in 0.5 time units and a 50% chance of occurring in the parent which will not reproduce again until it is a terminal cell (at least 3 generations). This produces a bimodal distribution for the mutant proportion (see Figure 3D). For the rest of the paper we will consider only the case where mutations occur in reproducing cells.

Until now, we have not examined how aging might affect the spread of a growth mutation in budding and binary fission populations. Based on experimental studies we expect cells to reproduce more slowly as they age [50]. If a beneficial mutation occurs in a new cell in the population, then a reduction in the growth rate for older cells should benefit the spread of the mutant by giving it a larger relative growth advantage compared to the older, ancestral population. Since budding and binary fission partition age differently across cells (see Figure 1), we may expect differences between them in terms of how the mutant increases.

In line with our expectations, Figure 4 and Figure 5 show that age-related fitness costs increase the mutant proportion in all populations. In the case of binary fission (Figure 4), there is no difference between how cells reproduce when they are unicellular or as members of a multicellular filament. The distributions have more dispersion than those without age-related fitness costs (Figure 3) because the growth rates of ancestral (or mutant) cells are no longer identical. The different ages across cells translates to more variation among reproductive times which causes dispersion in the distribution. We see this effect in Supplemental Figure A1 where we compare the spread of the mutation in populations with a small aging factor (1.025) to those with no age-related fitness costs but instead a Gaussian noise term added to the reproductive times. We also note that the reason the mutation has a higher fold increase in populations without age-related fitness costs is because the population is perfectly synchronized—addition of a noise term shifts the distribution so that age-related fitness costs have a monotonic effect of increasing the rate the mutant spreads.

If age is defined as the average age of poles, rather than the oldest pole, we find that it takes a higher value of the aging factor to alter the spread of the mutant in the population (see Figure A2). By comparing the two formulations of age in binary fission populations, we see that the population with the more asymmetric partitioning of age (maximum pole age) has a higher final proportion of the growth-rate mutant (compare aging factor = 1.1 for Figure 4 and Figure A2).

Unlike binary fission populations, unicellular and multicellular budding populations are not equivalent. Every cell in a unicellular budding population can reproduce so in the absence of age-related fitness costs, unicellular budding populations are identical to binary fission populations (either unicellular or multicellular). However, age-related fitness costs cause unicellular budding populations to differ from binary fission populations in terms of adaptation (see Figure 5). Impairing the reproductive rate of older cells still increases the spread of the mutation but not by the same extent as in binary fission populations (average fold increase of 79.18±5.64 in unicellular budding versus 131.21±15.29 in binary fission populations for aging factor 1.1). It is unclear though if such comparisons are fair because the way age accumulates is different between binary fission and budding populations. If we instead compare unicellular budding populations to binary fission populations where age is defined as the average age of poles rather than the oldest pole (see Figure A2), we find that unicellular budding populations show the higher relative increase in mutant proportion.

Unicellular and multicellular budding populations are affected differently by age-related fitness costs. In Figure 5, we see that that the distributions of the mutant fold increase have different shapes: unimodal in unicellular budding populations and bimodal in multicellular budding populations. The shape differences are due to the limitations on which cells can reproduce in multicellular filaments. Another difference is that aging can enhance the spread of a mutant by a much larger factor in multicellular populations compared to unicellular populations (216.40±0.82 in the higher mode of multicellular populations for aging factor 1.1 compared to 79.18±5.64 for unicellular populations). Indeed, if a mutation lands in a terminal cell its fold increase is higher in budding populations with an aging factor of 1.05 than the highest fold increase in a unicellular budding population (or a multicellular binary fission population) with an aging factor of 1.1.

The limitation concerning which cells can reproduce in multicellular budding populations causes them to take longer to reach carrying capacity than either unicellular budding populations or populations reproducing via binary fission. Such a reduction in growth rate represents a significant fitness cost, making multicellular budding populations less competitive against other populations. Drawing inspiration from extant budding multicellular organisms that retain cytoplasmic continuity between daughter cells, we consider the possibility that cells in a filament cooperate by sharing resources. Specifically, if non-reproducing cells share resources with reproducing cells it could help them reproduce faster on average than if they were isolated cells, mitigating their growth rate deficiency. We implement resource sharing in our model so that budding multicellular populations reach their carrying capacity at the same time as populations reproducing via binary fission (see Appendix F for a comparison of the age structure in different populations). Figure 6 shows that resource sharing significantly increases the final proportion of a growth-rate mutant, with the fastest spreading mutants experiencing an average fold increase of 341.17±5.7 which is ≈3 times higher than binary fission populations (see Figure 3 for comparison).

## 4. Discussion

In this paper, we examine the spread of beneficial mutants (faster growth) in both unicellular and multicellular populations. By considering a mutation that has comparable effects on organisms in either unicellular or multicellular states, our approach allows us to examine the impact of multicellularity on adaptive processes. Specifically, we focus on examining how spatial constraints imposed by a simple multicellular growth form and variation in the cell’s mechanism of reproduction (i.e., budding or binary fission) affect the rate a beneficial mutation spreads. This comparison is useful because multicellular organisms necessarily evolve from unicellular ancestors, and must somehow outcompete single-celled organisms in order to persist. While this is generally thought to require selection favoring increased size (obtained via multicellularity, see p. 67 of [56] for an excellent summary of the argument), the evolutionary consequences of multicellular morphology are also consequential. Our results show that beneficial mutations spread faster in budding multicellular populations than their unicellular counterparts, though this depends on whether mutations arise in newly created cells or the parents of those cells.

Adaptation in nascent multicellular populations likely involves signatures of evolutionary history [57,58]. For instance, the form of division of labor amongst cells inside a multicellular organism may depend on the form of temporal plasticity that existed in its unicellular ancestors [22,26,27,28]. Our work illustrates an interesting case of contingent evolution. With regards to the rate of spread of a beneficial mutant, unicellular organisms that reproduce through budding are not predicted to differ from those that reproduce by binary fission. However, a difference that is neutral in the unicellular context becomes substantial in the multicellular context, affecting the rate and extent of the spread of beneficial mutations. Furthermore, other cell-level features, such as aging, interact in new ways with cell reproductive mode following the evolution of multicellularity. In this way, a major evolutionary transition can dramatically amplify subtle differences in lower-level units, affecting both the tempo and mode of subsequent evolution of higher-level units in a contingent fashion.

The reason budding in multicellular populations has a different pattern of adaptation from the other populations considered is that the ability for cells to reproduce depends on their location in the organism. Filaments represent a severe form of this limitation; in order to maintain the filamentous structure only the two outermost cells in each filament can reproduce (via budding). Other multicellular organisms composed of cells that reproduce via budding are likely to be less restrictive. For example, in snowflake yeast, budding cells form branching multicellular structures [29,59]. Within these structures cells can reproduce several times until they run out of physical space or nutrients and stop reproducing [11]. By the time this has happened these cells tend to be centrally located within the group. So there are still limitations correlated with location, but they are less restrictive than those in filamentous multicellularity. Interestingly snowflake yeast share a feature similar to the linear filaments studied here in that non-reproductive internal cells can regain the ability to reproduce again after the multicellular group fragments and they are no longer spatially constrained.

There is no unequivocal way to measure the effects of aging—both chronological time and number of replicative cycles can drive senescence [49,60]. Measures of aging based on each of these metrics are termed chronological and replicative age, respectively. In this paper, we focused on chronological age, as this measure is more straightforward to interpret and is used more widely than replicative age. We would not expect this choice to qualitatively affect our conclusions in either unicellular populations or multicellular filaments growing with binary fission, because every cell can reproduce at every time step. As a result, chronological age and replicative age should be highly correlated. We would, however, expect differences in multicellular filaments that grow via budding, since cells experience long periods in between reproduction events.

While we have focused our results on the evolution of nascent multicellularity, they should apply to spatially-structured growth more generally. Microbial growth is often limited to surfaces (e.g., the edge of an expanding biofilm), such that inner cells cannot divide [61,62]. These colonies should act much like budding filaments, in that nearly all the growth occurs on just the outermost cells, which our work suggests should accelerate the spread of beneficial mutations. Indeed, prior work on microbial colonies has shown that spatially-structured population growth allows beneficial mutations to reach much higher frequencies than when populations grow by uniform expansion in well-mixed liquid culture [63,64,65]. Furthermore, the accelerated rate of increase is highly sensitive to the location of the mutant cell within the expanding colony—mutations that occur at the growing frontier establish and rapidly increase in frequency while those that occur towards the interior becoming trapped as non-growing ‘bubbles’ [65,66]. Since 40–80% of all bacteria and archaea on Earth live in biofilms [67], many of which will be diffusion-limited, understanding how topologically-structured growth affects evolutionary dynamics stands to be quite consequential.

## Figures and Tables

**Figure 1 genes-12-00661-f001:**
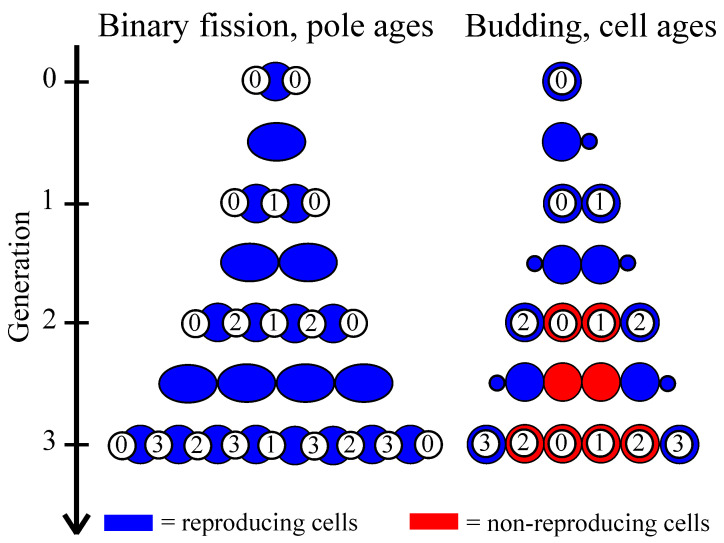
**Schematic for cell reproduction via binary fission versus budding in multicellular filaments.** Multicellular filaments can arise from cells reproducing via binary fission (**left**) or budding (**right**). In binary fission, cells (blue ovals) increase in size and then split down the middle to generate two daughter cells. Each daughter cell gets a newly synthesized pole and one from its parent (white circle with number indicating when it originated). The final age distribution of poles has a characteristic pattern where young and old cell poles are next to one another and the oldest poles are at the ends of the filament. In budding (right) each cell generates a bud that becomes a daughter cell. When budding cells form a multicellular filament this pattern of growth means that only terminal cells can reproduce (non-reproducing cells shown in red). Cell age is then the difference between the current generation and when the cell was created (number in white circle), and so the youngest cells are at the ends of the filament.

**Figure 2 genes-12-00661-f002:**
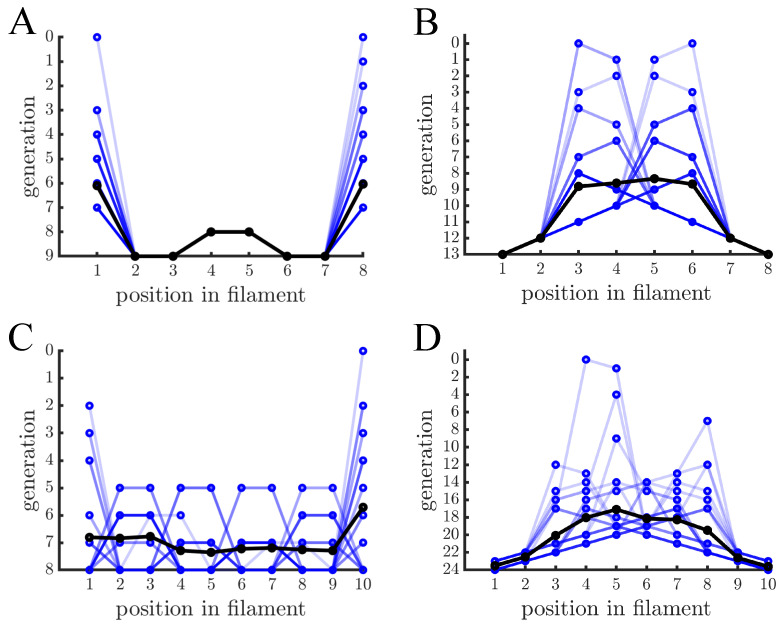
**Age distribution within multicellular filaments.** (**A**) The age of cells within multicellular filaments is shown as a function of cell position. Cells reproduce via binary fission and filaments reproduce once they reach 8 cells— different blue lines indicate different multicellular filaments and black lines indicate the population mean. A cell’s age corresponds to the generation that its oldest pole was created. In this case the oldest cells are at the ends of the filament and the next oldest cells are in the middle. (**B**) The graph is similar in structure to (**A**) except cells reproduce via budding so cell age is the generation in which the cell was created. Contrary to binary fission, the ends of the filaments contain the youngest cells and the older cells are in the middle. (**C**,**D**) The graphs are similar to (**A**,**B**), respectively, except that multicellular filaments reproduce at 10 cells. Budding has the same basic age structure but binary fission has a less regular pattern because not every cell has reproduced (i.e., filament size is not a power of 2).

**Figure 3 genes-12-00661-f003:**
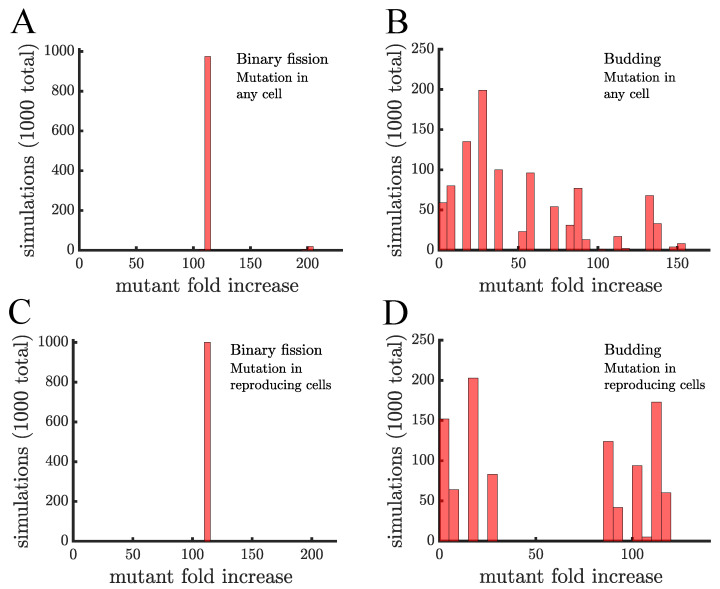
**The spread of a fast-growing mutant without age-related fitness costs.** (**A**) A histogram shows the fold increase of the mutant fraction ($\frac{\mathrm{final}\;\mathrm{fraction}}{\mathrm{initial}\;\mathrm{fraction}}$, where the initial fraction is $10^{-3}$) for $10^{3}$ independent simulations of a population reproducing via binary fission. In this panel the mutation is introduced in a random cell from the entire population regardless of when it reproduces. There is almost no dispersion in the histogram because at the point the mutation is introduced every cell is identical. The only outlier bar comes from the case in which the mutation is introduced in a cell that is just about to reproduce, i.e., the mutation is introduced when the population reaches 1000 but by the end of that time step the population will be 1024. (**B**) This panel shows the same type of data as a) but for a population of cells reproducing via budding. Here, cell reproduction depends on location in the filament so at the point the mutant is introduced cells differ in terms of their next expected reproduction time. The histogram is much more dispersed and the majority of simulations (868) have a lower mutant fold increase than in populations reproducing via binary fission in (**A**). (**C**,**D**) These panels are similar to (**A**,**B**), respectively, except the mutant is introduced in cells that just reproduced. This has no effect on binary fission populations but in budding populations it produces a more bimodal distribution with 498 mutations having a fold increase >75 and 502 having a fold increase <30, depending on whether the mutation occurs in the daughter or parent cell, respectively.

**Figure 4 genes-12-00661-f004:**
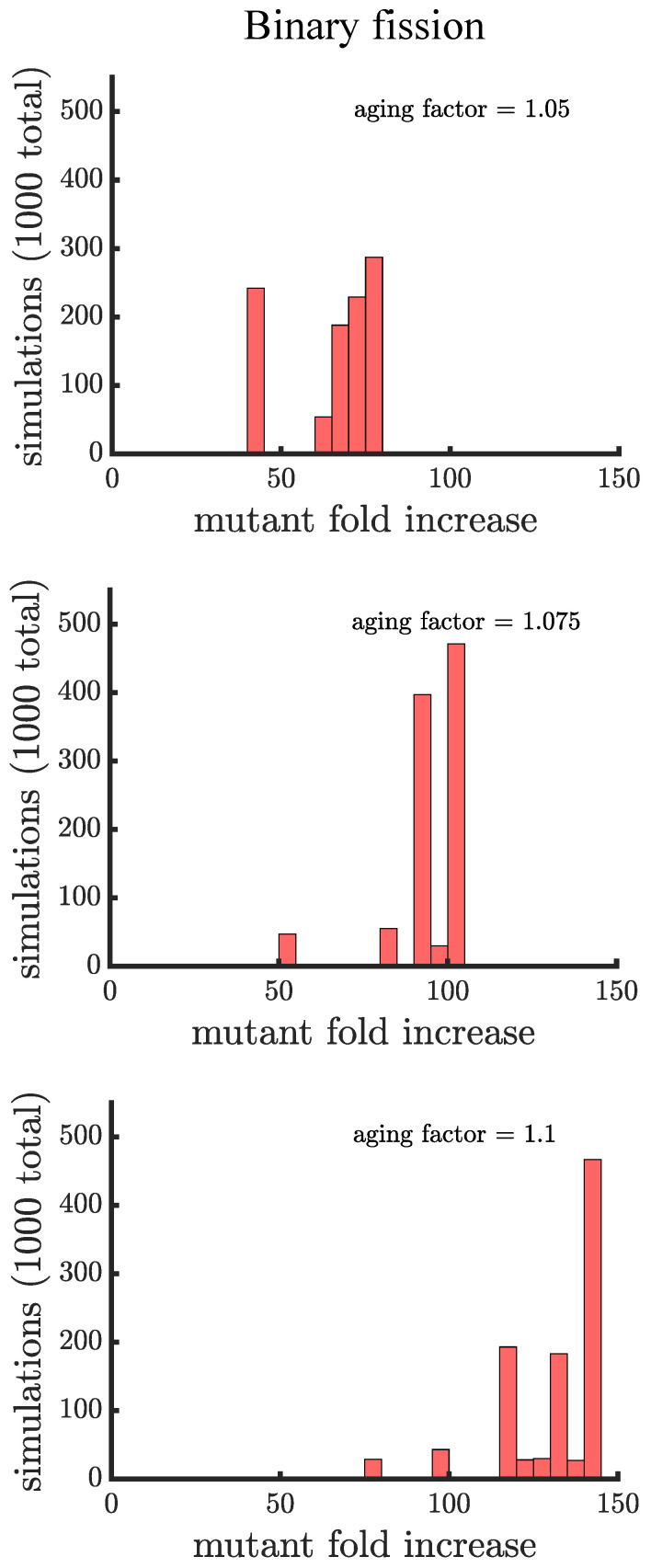
**The spread of a fast-growing mutant in binary fission populations with age-related fitness costs.** Plotted are histograms of the fold increase of the mutant fraction ($\frac{\mathrm{final}\;\mathrm{fraction}}{\mathrm{initial}\;\mathrm{fraction}}$, where the initial fraction is $10^{-3}$) for $10^{3}$ independent simulations experiencing different age-related fitness costs. The aging factor (the parameter *b* in Equation (Equation 1)) is the multiplicative factor by which reproductive time increases with age such that 1.05 means a 5% increase in reproductive time in the simulations. As the aging factor is increased the distribution shifts to the right indicating that the final mutant fraction increases in simulations. See Appendix C, Appendix C and Appendix C for robustness analyses and companion figures using different simulation parameters.

**Figure 5 genes-12-00661-f005:**
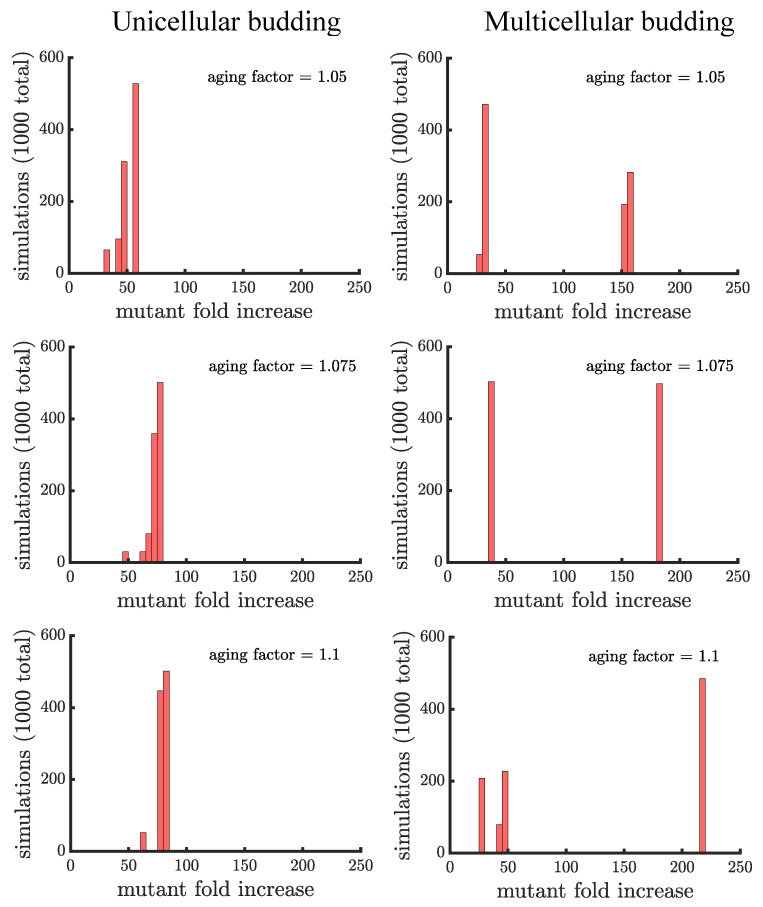
**The spread of a fast-growing mutant in budding populations with age-related fitness costs.** (**left**) Plotted are histograms of the fold increase of the mutant fraction ($\frac{\mathrm{final}\;\mathrm{fraction}}{\mathrm{initial}\;\mathrm{fraction}}$, where the initial fraction is $10^{-3}$) for $10^{3}$ independent simulations of unicellular budding populations experiencing different age-related fitness costs (similar in structure to Figure 4). Increasing the aging factor shifts the distribution to the right elevating the final mutant fraction. (**right**) Plotted are another set of histograms similar to the left panel except for multicellular budding populations. Compared to the unicellular budding populations, the distributions are more bimodal. If we consider the higher mode of the distributions we find that again the aging factor increases the final mutant fraction but by a much greater amount than in the corresponding unicellular populations: 216.40±0.82 in a multicellular budding population for an aging factor of 1.1 compared to 79.18±5.64 in the unicellular budding population. See Appendix C, Appendix C and Appendix C for robustness analyses and companion figures using different simulation parameters.

**Figure 6 genes-12-00661-f006:**
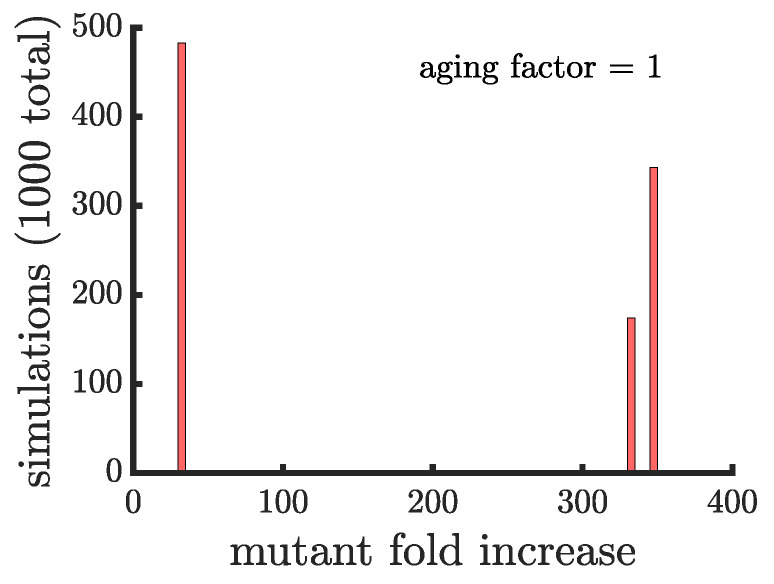
**The spread of a fast-growing mutation in multicellular budding populations with resource sharing.** The histogram shows the fold increase of the mutant fraction ($\frac{\mathrm{final}\;\mathrm{fraction}}{\mathrm{initial}\;\mathrm{fraction}}$, where the initial fraction is $10^{-3}$) for $10^{3}$ independent simulations of multicellular budding populations that have resource sharing ($k_{0}=0.46$) and no age-related fitness costs. The budding populations reach the carrying capacity at the same time as a population reproducing by binary fission. The distribution is still bimodal but the higher part of the distribution has increased by a factor of ≈3 compared to Figure 3.

## Data Availability

Not applicable.

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
