# Peer review of "The Consequences of Budding versus Binary Fission on Adaptation and Aging in Primitive Multicellularity"

_genes, 2021, doi:10.3390/genes12050661_

Round 1

Reviewer 1 Report

This paper is clear, well written, and makes a strong case on the effect of different forms of development in evolutionary processes. In particular, the paper evaluates how different forms of cell-division can affect the spread beneficial mutations, in single cell populations and their multicellular counterparts. Two main factors affect the spread of mutations in budding vs. binary fission multicellular populations: which cells get to reproduce (with only edge cells reproducing in binary fission), and the age distribution of cells (which is modeled in this paper as a cost on growth).

Overall I thought the paper was really clear and I really enjoyed reading it, but I think the difference of these two factors could be made clearer. For example, what happens in cases where the age structure is different, but all cells can reproduce. Or more likely to happen in nature, what if not all of the cells in a filament reproduce?

My main other comment is with regards to the framing and structure of the discussion. I wonder if it is more powerful (and potentially more directly linked to this paper) to emphasize how differences in development (and cellular traits) can have consequences later on for the evolution and adaptation of multicellular organisms, instead of an emphasis on conflict and the relative fitness of multicellular organisms with respect to their unicellular ancestor. At the end, this paper is not about the origins of multicellularity, nor the relative fitness benefit of multicellular organisms and their unicellular ancestor. It is instead, about the evolutionary consequences of different morphologies and forms of cell reproduction.

Along these lines, I found the paragraph on “fitness decoupling” unnecessary and distracting. I think the idea of fitness decoupling is really interested and I have really enjoyed previous papers of these and other authors on the subject, but I do not think it contributes much here and can confuse readers less familiar with the literature. Moreover, fitness decoupling is interesting when there is a trade-off between fitness of single-cells and fitness of cells in a multicellular organism. But in the models presented here (in the absence of division of labor), it is not clear that such trade-off is particularly relevant (it might be in the budding case, where not all cells reproduce, but again that is not the focus of the paper). I have also wondered about cases in which fitness at both levels is aligned but for some reason selection is stronger at the multicellular level, favoring multicellular level adaptations (e.g. there is low variation/availability of mutations increasing fitness at the cellular level, but multicellular level adaptations are readily available).

Minor comments

Line 20 - (This is mostly out of curiosity for me) I was intrigued on why did you choose to cite the Evolution paper of Rebolleda-Gomez and Travisano and not our paper in AmNat? In my mind, the second paper is more relevant to this paper, it shows how growth, morphology and spatial structure can shape the fitness costs of unicellular its vs. multicellularity. 

L. 44-47 - I think the discussion of fitness decoupling is not particularly relevant in this paper and it cuts a little the flow of the introduction. 

L. 160 (and that whole section) - It would seem to me that resource sharing could affect more than just the time to carrying capacity. Wouldn’t it also spread the cost/benefits of growth differences across cells? I wonder if there is some other normalization that you could use to account for the differences to carrying capacity. 

L. 235 - Are these simulations computationally expensive or slow? It feels like 50 simulations lead to fairly sparse distributions, and I wondered if better sampling would provide additional information of relevance (like the relative heights of the two main peaks in the bimodal distributions and if that is something that changes with increased aging costs). 

L. 267 - Show in Figure 4 the unicellular case (just for completion). 

Reviewer 2 Report

I recommend this manuscript to be revised from the two points:

  1. This is not a very important point for the paper itself. The authors mention the "fitness decoupling" model. It is fine as it is part of the literature and historically important. I am quite critical of this model and I believe it would be beneficial, when citing the fitness decoupling model, to cite a few papers criticising the model (metaphorical nature of the model).  This would have no impact on the results of the paper.
  2. I think the explanations of Figure 3 and the followings could be improved. Spend a bit more time explaining what these figures are representing exactly. I understood, but I had to re-read it.

Other than that, the paper is clear.

Reviewer 3 Report

The paper by Isaksson et al presents an interesting approach to investigate the tempo and modo of multicellularity adaptation by means of a series of in-silico computational experiments. With their investigation, authors find that (i) the spread of mutations in multicellular budding populations is qualitatively distinct from the populations of cells duplicating by fission (in general slower) and that (ii) when imposing a growth rate that decreases with cell age, beneficial mutations are observed to spread significantly faster in a multicellular budding population than its corresponding unicellular population or a population reproducing via binary fission.

I find this article very interesting. Not only for the biological implications of the main findings, which are shedding a light on evolutionary mechanisms that led to the emergence of multicellularity, but also because it offers some insights into the investigation of spatially-structured evolving populations, which has recently became a hot topic in other research fields (e.g., cancer evolution).

I would ask for the following minor revisions, which, in my opinion, would improve the support to the main conclusions of the paper. Indeed, since all the results are based on a computational analysis, authors should check the robustness of their findings against variations of key model parameters, focussing specifically on

  • Growth rate of the mutant cells. At the moment this is set to twice the growth rate of the ancestral cells. Authors should vary this difference in a wider range (both lower and bigger than two fold) and see how their results depend on this parameter.
  • Population size when the mutation is introduced. At the moment this is arbitrary set to 10^3. Again, authors should vary this parameter, for example in the range [10^2,10^4]

Other comments:

  • Fig. 2: authors could add a shadowed area to visualize the mean +/- standard deviation evaluated over many replicates. This would help the reader to capture the global trend at a glance. 
